# Bronchial Artery Embolisation in Haemoptysis Management: A Scoping Review with Emphasis on Embolic Materials and Indications

**DOI:** 10.3390/arm93050035

**Published:** 2025-09-12

**Authors:** Anna Ziętarska, Adam Dobek, Anna Sawina, Piotr Białek, Sebastian Majewski, Ludomir Stefańczyk

**Affiliations:** 1I Department of Radiology and Diagnostic Imaging, Norbert Barlicki Memorial Teaching Hospital No. 1, Medical University of Lodz, 90-153 Lodz, Poland; anna.zietarska@stud.umed.lodz.pl (A.Z.); anna.sawina@student.umed.lodz.pl (A.S.); piotr.bialek@umed.lodz.pl (P.B.); ludomir.stefanczyk@umed.lodz.pl (L.S.); 2Department of Pneumology, Norbert Barlicki Memorial Teaching Hospital No. 1, Medical University of Lodz, 90-153 Lodz, Poland; sebastian.majewski@umed.lodz.pl

**Keywords:** haemoptysis, bronchial artery embolisation (BAE), embolic materials, polyvinyl alcohol (PVA), N-butyl-2-cyanoacrylate (NBCA), coils, gelatin sponges (GS), microspheres

## Abstract

**Highlights:**

**What are the main findings?**
Bronchial artery embolisation (BAE) remains a versatile therapeutic option for haemoptysis, with a wide range of embolic materials currently in use.Polyvinyl alcohol (PVA), coils, gelatin sponges (GS), and N-butyl-2-cyanoacrylate (NBCA) have distinct profiles that influence their clinical use.

**What is the implication of the main finding?**
Clinical decision-making in BAE should be tailored to the patient’s condition and embolic material properties.The development of standardised management guidelines for patients requiring bronchial artery embolisation is essential to improve outcomes.

**Abstract:**

Haemoptysis is an alarming symptom of a wide spectrum of underlying diseases, ranging from indolent chronic conditions to life-threatening states. Among the strategies to manage pulmonary bleeding is bronchial artery embolisation (BAE), an interventional radiology procedure. The objective of this scoping review was to map the current evidence on embolic agents used in BAE for haemoptysis management, with a focus on their clinical applications, and decision-making factors. Studies published between 2019 and 2024 were included if they specified the embolic material used and reported outcomes of BAE in adult patients. Data were extracted from PubMed and charted according to embolic agent type, recurrence rate, and clinical context. Thirty-one studies met the eligibility criteria. Polyvinyl alcohol (PVA) remains the most widely studied agent, comparable in efficacy to more homogeneous microspheres. Gelatin sponges (GS), though biodegradable, are well-documented and affordable, making them a common choice. N-butyl-2-cyanoacrylate (NBCA) is highly effective for small vessels and may offer lower recurrence rates. Coils are valuable in proximal embolisation and severe cases. This review highlights the need for individualised embolisation strategies and updated guidelines for material selection, considering clinical context, vascular anatomy, and recurrence rates. The findings aim to support evidence-based decision-making in interventional radiology practice.

## 1. Introduction

Haemoptysis, while a relatively rare symptom, often signals serious underlying conditions and poses significant diagnostic and therapeutic challenges. It refers to the expectoration of blood or blood-streaked sputum originating from different parts of the respiratory tract beneath the level of the larynx [1,2]. The most common causes include active tuberculosis and its sequelae, such as bronchiectasis or aspergilloma [3]. Other notable causes are cystic fibrosis and lung tumours.

In most cases, bronchial arteries (BAs) are the primary source of bleeding in haemoptysis. However, non-bronchial systemic arteries (NBSAs) can also contribute. Among these, the internal thoracic, intercostal, and inferior phrenic arteries are the most frequently involved [4,5].

Based on the volume of expectorated blood, haemoptysis is often classified as either massive or non-massive (mild or moderate) [1]. This classification frequently influences the choice of treatment. Non-massive haemoptysis is typically managed conservatively or with vasoactive medications, whereas massive haemoptysis requires more invasive approaches, such as bronchoscopy, bronchial artery embolisation (BAE), or surgical intervention [6].

Nevertheless, the clinicians are not unanimous about the threshold distinguishing massive from non-massive haemoptysis. According to different interpretations, the amount of blood varies between 100 mL and 1000 mL per day [2]. The lack of a definitive threshold significantly hinders research interpretation and potentially influences the clinical application spectrum of bronchial artery embolisation (BAE).

BAE is a well-established method for controlling massive or recurrent haemoptysis with proven safety and high efficiency [7]. While traditionally considered an emergency intervention for massive haemoptysis, BAE also serves a palliative role, improving the quality of life in patients with chronic conditions where surgery is not an option [8,9,10,11].

However, the high recurrence rate remains a problem. The reported recurrence rates vary between the studies, due to multiple factors, including diameter and localisation of the culprit vessel, administration of vasoconstrictors, and different underlying conditions [6].

Despite its growing application, there are no explicit guidelines concerning this procedure itself, including recommendations for specific embolic materials.

This scoping review aims to map the existing evidence on embolic agents used in BAE for haemoptysis, identify patterns in clinical application, and highlight gaps in current practice and research. Given the heterogeneity of study designs, definitions, and outcome measures, a scoping review was chosen to provide a comprehensive overview rather than a quantitative synthesis. The review focuses on studies published in the last five years that report clinical outcomes of BAE in adult patients, with specific reference to the embolic materials used.

This review explores the key embolisation agents used in BAE for the management of haemoptysis, highlighting their advantages, limitations, and reported clinical applications. By synthesising the available data, it aims to support clinical decision-making and guide future research directions.

## 2. Materials and Methods

### 2.1. Search Strategy

A literature search was conducted in the PubMed database on December 20, 2024, using the following query: (hemoptysis[Title/Abstract]) AND (embolization[Title/Abstract]).

Filters were applied to include full-text articles published in English between 2019 and 2024, involving human subjects. The search yielded 199 results across 20 pages. The aim was to map the existing literature on bronchial artery embolisation (BAE) for haemoptysis, with a focus on embolic materials used and clinical outcomes.

Studies were screened manually by one reviewer. Articles were excluded if they were not relevant to the topic or were single-case reports.

The final review encompassed 31 studies identified through the structured selection process (Figure 1).

Although some studies were excluded from quantitative synthesis due to lack of extractable data or atypical focus, they were retained for narrative context where appropriate. This approach reflects the scoping nature of the review and supports a comprehensive understanding of embolisation in haemoptysis. Additional publications were included as a background, based on their relevance, citation frequency, or value in illustrating temporal trends in the use of embolic material. Details regarding the characteristics of the included studies are presented in Table A1, located in Appendix A. Data were extracted and reported following the PRISMA-ScR guidelines, including study authorship, year, patient population, underlying aetiologies of haemoptysis, embolic materials applied, and outcomes such as recurrence rates during follow-up [13]. Although no formal protocol was registered, all methodological steps were documented and reported in line with the PRISMA-ScR recommendations to ensure thematic relevance and data accessibility.

### 2.2. Eligibility Criteria

The inclusion criteria listed below were applied during the selection process.

Reports specifying the embolisation agent(s) used (key criterion);Studies focusing on embolisation for haemoptysis using conventional embolic agents (NBCA, coils, PVA, microspheres, gelatin sponges);Outcomes measured as clinical success (immediate haemoptysis control), haemoptysis-free survival, or recurrence rate;Original studies published between 2019 and 2024;Articles handling typical haemoptysis aetiologies such as bronchiectasis, tuberculosis, cystic fibrosis, aspergillosis or malignancy;Studies available in full text in English.

Studies that met any of the exclusion criteria outlined below were removed from further consideration.

Studies not reporting the type or frequency of embolic agent used (key criterion);Publications focusing primarily on atypical causes of haemoptysis not listed among the inclusion aetiologies;Chemoembolisation or embolisation involving therapeutic (non-occlusive) agents or procedures;Articles published outside the 2019–2024 timeframe;Editorials, letters, commentaries, or non-peer-reviewed documents;Studies focused solely on non-bronchial systemic embolisation;Animal or in vitro studies;Studies not available in full-text form.

While reasons for exclusion were clearly defined, many records met multiple criteria simultaneously. Therefore, individual exclusion categories were not quantified separately, in order to maintain clarity and avoid unnecessary granularity.

Studies were eligible regardless of the severity of haemoptysis reported. The definition of massive haemoptysis varied widely across the studies included in this review, with thresholds ranging from >100 mL to >600 mL per 24 h, and some incorporating clinical criteria such as respiratory failure or haemodynamic instability (Table A2). Due to this heterogeneity and the imprecision of volume estimation, this review did not classify haemoptysis severity based on volume. Where terms such as “massive” or “non-massive” haemoptysis are used in this review, they reflect the original classification applied by the respective study authors.

Definitions of complications differed between studies; therefore, to standardise reporting, we summed major and minor events into one overall complication rate, noting that some minor events (e.g., transient chest pain) were not consistently documented. Clinical success was defined in line with the majority of included studies as immediate haemoptysis control, that is, cessation of bleeding within 24 h of BAE. Aggregated values are presented in the main text, while the detailed, study-level source data are provided in Appendix A (Table A3).

### 2.3. Study Selection and Data Charting

Data were extracted using a pre-designed Excel form that included the following variables: year, study type, patient population, embolic material used, definition of clinical success, reported complications, and recurrence outcomes. Extraction was performed by a single reviewer, with any uncertainties resolved through consultation with a second researcher to ensure consistency and accuracy.

The extracted data were synthesised both descriptively and in tabular format. Results were grouped according to the type of embolic material to facilitate comparative analysis.

## 3. Polyvinyl Alcohol (PVA)

### 3.1. Short Description

Polyvinyl alcohol (PVA) belongs to a group of particulate embolic agents, along with gelatin sponges. These materials have been the most widely used for many years because of their low cost, availability, and customizable fit, and they have well-documented safety in haemoptysis control [14]. The fact that PVA particles can cause permanent embolisation gives them an edge over resorbable gelatin sponges. PVA may be used as the only embolisation agent or in combination with others during BAE. PVA particle dimensions of 300–500 µm are most frequently used [3]. Reported rates of technical and clinical success range from 93.9 to 100% and 91.7 to 100%, respectively. The one-year haemoptysis recurrence rate in these studies varied between 9.7% and 23% [14,15,16,17]. However, these values are difficult to interpret due to the significant heterogeneity in participant numbers and clinical scenarios between trials.

### 3.2. Advantages

PVA particles are characterised by their good histocompatibility and non-biodegradability, which provides durable occlusion. Patients tolerate this material well with low rates of serious complications. The complication rate of PVA particles is comparable to that of NBCA or microspheres in BAE [17,18].

Due to the availability of PVA particles in various sizes (150–1200 µm), they can be utilised to embolise vessels with different diameters [3]. In practice, particles sized 300–500 µm are most often selected, as they help prevent both non-target embolisation and are more proximal than intended embolisation [15]. Small PVA particles can reach more distal vessel segments, which is relevant for preventing recurrence. Sheehan et al. achieved good outcomes using even smaller particles with a diameter of 150–250 µm. They did not report complications associated with non-target embolisation such as lung stroke, which authors attributed to meticulous technique during BAE, including frequent use of diagnostic arteriograms [15]. Avoiding wedge catheter positions is considered crucial for clinical success.

### 3.3. Disadvantages

Despite their undeniable advantages, the use of PVA is not devoid of shortcomings. When released into the bloodstream from a microcatheter, particles expand and occlude the targeted vessel [17,18,19]. Thanks to their good expansibility, particles can permanently occlude the intended vessel. Nevertheless, they are prone to recanalisation due to poor adhesion to the endothelium [19]. This may pose a challenge given that pulmonary artery diameter fluctuates with changes in blood pressure, both physiologically and due to various pathologies. Vasoconstrictor therapy works similarly. Using vasoconstrictors such as terlipressin up to 5 h before the BAE procedure poses a risk of incomplete embolisation [20].

Additionally, the varying size of PVA particles promotes aggregation within the catheter, leading to flow blockage and, potentially, premature embolisation [3,14,17,18]. Ectopic embolisation is another potential complication associated with polyvinyl alcohol, particularly when particle sizes are below 300 µm or in the presence of shunts [3,15,20]. Hanotin et al. did not observe it in their study but concluded it is a real danger in patients with large volumes of intercostal pulmonary venous shunt and high blood and injection pressure [20]. The concurrent use of coils during the procedure can reduce blood flow and presumably prevent ectopic embolisation. However, the number of patients with this rare vascular anomaly was limited, so it is hard to extrapolate the obtained results to a wider population of patients with the presence of shunts who suffer from haemoptysis.

Special care is also essential during BAE with using small particles of PVA (150–250 µm). In this study, Sheehan et al. reported promising results regarding the application of smaller PVA particles in practice. However, in 33 out of 171 procedures, these particles were insufficient to occlude the vessel, making it necessary to continue embolisation with larger particles [15]. It would be valuable to explore the efficacy of controlling haemoptysis with small PVA particles depending on the volume of bleeding in this study, an aspect which has not been analysed. The risk of severe complications after ectopic embolisation may be overestimated in present times, but should not be neglected. Underlying disease, intensity of haemoptysis or anatomical conditions, and frequent use of diagnostic arteriograms may influence the outcome.

Another notable drawback of polyvinyl alcohol is its radiolucency, which increases the risk of accidental embolisation [19]. To ensure successful embolisation, PVA particles must be combined with contrast. The BAE procedure using PVA lasts longer than the analogic technique with n-butyl-2-cyanoacrylate, so the radiation dose is also proportionately increased [21]. Furthermore, repeated arteriograms, as proposed by Sheehan et al., increase radiation exposure, raising the question of whether using 150–250 µm PVA particles justifies the associated risk [15].

All these factors, including poor adhesion, variability in particle size and promoting aggregation, risk of ectopic or unintended proximal embolisation, and radiolucency of PVA, may contribute to a higher recurrence rate.

Lee et al. reported that the majority of recurrences after BAE using PVA occur within the first month following the procedure, and the use of PVA itself is an independent factor in the recurrence of haemoptysis [21]. The recurrence within this short period is most commonly associated with recanalisation or incomplete embolisation of the vessel. Among the studies considered in this review, the average haemoptysis-free survival rates were 77–91.1% in the first year, 75–79% in the second year after BAE, and only 66% after five years. As a result, the long-term efficacy of embolisation procedures using PVA may be limited, giving way to more efficient materials [14,15,21,22].

### 3.4. Clinical Use

PVA is widely used in bronchial artery embolisation (BAE), alone or with other agents, for cases ranging from massive haemoptysis to palliative care in chronic lung disease. Fu et al. embolised intercostal arteries with PVA particles, alone or with coils, in patients with massive haemoptysis due to bronchiectasis, TB, etc. [16]. Beyond major bleeds, Sheehan et al. support BAE with small PVA particles in persistent non-massive haemoptysis [15]. Cystic fibrosis-related massive haemoptysis can be treated via BAE or non-BAE methods [23]. Palliative embolisation (e.g., for CF or cancer) offers good tolerance and rapid relief. Fan et al. showed BAE (mostly using PVA) outperformed conservative therapy [24]. Early intervention lowers recurrence and improves hospitalisation time and activity return [25]. These results support BAE’s growing role in palliative/conservative treatment. Still, PVA lacks the durability of n-butyl-2-cyanoacrylate. Lee et al. found PVA predicted recurrent haemoptysis. In primary lung cancer, NBCA was more effective due to vessel morphology; PVA struggles in small, tumour-fed vessels [21]. Woo et al. found NBCA was more effective than PVA in the treatment of haemoptysis associated with bronchiectasis. However, in cases of tuberculosis and aspergilloma, there were no significant differences in haemoptysis-free survival periods between these two embolic agents, likely due to the progressive neovascularisation [14].

In summary, although PVA remains a key embolisation agent, it requires case-specific use, as risk of recanalisation and recurrence demands individualised treatment strategies.

## 4. Microspheres

### 4.1. Short Description

Microspheres belong to the group of particulate embolic agents similar to polyvinyl alcohol. Calibrated microspheres have standardised size and spherical shape, which makes them more homogeneous than PVA. The diameter of microspheres commonly used in embolisation ranges from 100 to 1300 μm, although some may be as small as 40 μm [26,27,28,29]. Proper selection of microsphere size is crucial, as it ensures their accumulation at the target site within the selected vessel. However, numerous microsphere types vary in physical properties, including wettability and biodegradability. Additionally, they might have further features like thrombin or PEG coating. Microspheres might also contain medical preparations or radioactive isotopes [26].

A common feature of PVA and microspheres is permeability to radiation, which necessitates mixing them with contrast agents [26,29].

### 4.2. Advantages

Calibrated microspheres address the issue of aggregation caused by the irregular shape of other embolising agents in this group. Unlikely PVA particles, microspheres do not tend to clump and instead create a single-file alignment inside the vessel [26,27]. Consequently, they do not lead to microcatheter occlusion [27].

Successful embolisation depends on the deformation capability of microspheres, which determines their resistance to external forces and facilitates their passage through the microcatheter [26].

The availability of microspheres in a wide range of sizes allows precise targeting of the culprit vessel [26,27,29].

Microspheres offer multiple additional functions that may improve embolisation outcomes and enhance their capabilities. Although this review focuses primarily on using embolisation for haemoptysis cessation, it is worth noting that microspheres can also serve as a drug delivery system, directly targeting tumours. This approach, known as drug-eluting beads (DEBs), reduces systemic toxicity and induces apoptosis in tumour cells by occluding their arterial supply [26].

### 4.3. Disadvantages

Under the pressure, microspheres may fragment, increasing the risk of embolisation beyond the intended vessel and causing tissue necrosis [26].

To prevent bronchial necrosis, Mazıcan et al. used particles with a diameter of over 250 μm [27]. However, there is no clearly defined threshold for the minimal particle size that significantly reduces the risk of reflux while maintaining the effectiveness of BAE. Generally, using particles smaller than 300 μm for embolisation is not recommended due to the potential development of tissue necrosis. Nevertheless, selecting the appropriate particle size depends on the clinical context and the diameter of the target vessel [18,27,30].

Smaller microspheres reach more distal sites of the vascular beds and small-diameter vessels, while larger microspheres tend to embolise more proximally [31]. This distinction is crucial when selecting the appropriate microsphere size for a specific embolisation procedure, as it impacts both effectiveness and safety. Understanding the balance between occlusion level and particle size helps optimise therapeutic outcomes while minimising potential complications. To prevent non-target embolisation, especially in the anterior spine artery, when using microspheres smaller than 325 μm, delivering an embolising agent through a microcatheter as distally as possible is essential [27,29,31].

Microspheres may be composed of allogenic materials, potentially inducing foreign-body reactions or antibody synthesis. Particles ranging from 100 to 300 μm more frequently induce an inflammatory response than larger microspheres due to their increased tissue penetration [26].

Adding hydrophilic contrast to a particulate embolising agent increases its fluidity, facilitating movement within the vessel but also posing a risk of retrograde flow [26,29]. Simply adding contrast to microspheres does not ensure precise navigation of embolic material and may result in contrast particles escaping from the mixture, weakening the embolic structure. Free contrast particles might reach other organs and cause toxicity. Rapid contrast washout from the bloodstream necessitates repeated exposure to radiopaque substances and radiation [26].

### 4.4. Clinical Use

Though microspheres offer several improvements over PVA, both agents show similar efficacy in haemoptysis control, clinical outcomes, recurrence, mortality, and complications. In Fu et al., only haemoptysis-free survival differed—longer with PVA in non-bronchiectasis cases—though this may reflect small subgroup sizes. The study favoured microspheres due to lower cost [17]. Similarly, Xu et al. found reduced hospitalisation costs using 700–900 μm vs. 500–750 μm microspheres. Both sizes had comparable efficacy/safety, likely due to non-random selection and large diameters minimising non-target occlusion [31]. Still, multicentre, randomised trials are needed to standardise microsphere use based on anatomy and clinical context. Future studies should compare PVA vs. microsphere recurrence rates, actual frequencies, and underlying causes. Microspheres are also explored for palliative/pre-op treatment in lung tumours. For nonresectable, hypervascular tumours, arterial occlusion via small-diameter particles or NBCA is preferred [21,26,32]. Outcomes may improve by proximal embolisation with larger microspheres after distal occlusion [26]. Geevarghese et al. identified haemoptysis as a poor prognostic marker in primary/metastatic lung cancer, linked to disease progression and declining lung function [33]. Still, BAE was found to be safe, tolerable, and effective for symptom relief. Nomori et al. noted NBCA embolisation reduced growth in chemo-resistant, nonresectable tumours presenting with haemoptysis [32]. Comparing NBCA vs. staged microsphere embolisation could offer insights into conservative management of lung tumour–related haemoptysis.

Taken together, although microspheres are more homogeneous and standardised than PVA, both agents demonstrate comparable effectiveness in BAE. They also offer additional features; thus, their application goes beyond vascular occlusion, enabling broader therapeutic applications.

## 5. N-Butyl-2-Cyanoacrylate (NBCA)

### 5.1. Short Description

NBCA (N-butyl-2-cyanoacrylate) belongs to the group of non-particulate, liquid embolising agents. It was first used for embolisation of cerebral AVMs in the 1980s, although theories regarding its possible effectiveness had already emerged in the 1960s [5,34]. The most commonly used materials are Histoacryl (n-butyl-2-cyanoacrylate) (B. Braun Melsungen AG, Melsungen, Germany) and Glubran 2 (Gem S.r.l., Viareggio, Italy), which were introduced later, for strictly endovascular purposes [5]. N-butyl-2-cyanoacrylate is often compared to polyvinyl alcohol in terms of safety and efficacy. NBCA’s undeniable advantages include prompt vessel occlusion, a low recanalisation rate, the ability to control polymerisation time, and the capability to embolise even small-diameter vessels [14,21]. These features could provide glue embolisation a significant edge over PVA. However, despite the mentioned advantages, its use in bronchial artery embolisation remains limited.

Performing glue embolisation requires significant experience from the interventionist [5,18,27,28,35]. Improper handling of NBCA can lead to severe complications, including tissue necrosis or spinal cord infarction [3,6,28,36]. However, in the hands of qualified specialists, these sequelae rarely occur [5,14,21,27,28,35].

### 5.2. Advantages

NBCA embolises the offending vessel through a mechanism different from PVA. After exposure to tissues, NBCA polymerises and undergoes three stages: thrombotic material formation, adhesion to endothelium, and endothelial damage [5]. This makes NBCA a more permanent embolic material than PVA [14].

This process is independent of any defects in the patient’s coagulation pathway [21,37]. However, coagulation disorders typically affect only a small subset of patients, making direct comparisons between embolic agents challenging in such cases.

Additionally, NBCA is less viscid than PVA mixed with contrast [14]. This property allows NBCA to reach more distal parts of the vessels compared to PVA particles.

Woo et al. extensively investigated the use of NBCA in BAE. They emphasised a significant advantage of glue embolisation in preventing haemoptysis recurrence, which tends to occur earlier in patients embolised with PVA. This results in a reduced need for repeat embolisation of the same vessel. NBCA has a lower recanalisation rate compared to PVA, which Woo et al. attribute to its ability to reach more distal segments of the targeted vessel [14]. In contrast, Lee et al. explained this by NBCA’s specific mechanism of action, particularly polymerisation at occlusion sites [21].

The success of glue embolisation largely depends on the dilution of NBCA with Lipiodol [5]. Determining the appropriate ratio allows interventionists to control both the site of occlusion and the timing of polymerisation [35]. The optimal proportion depends on the vessel’s diameter and flow rate [21]. More diluted mixtures, such as NBCA-to-iodised oil ratios of 1:6 and 1:8, can reach more distal segments of the target vessel due to their liquidity and delayed polymerisation, making them suitable for embolising small-diameter vessels [5,28]. For large-diameter vessels with high blood flow, more concentrated mixtures (e.g., 1:3 or 1:2) are preferred to ensure rapid polymerisation [5].

The most commonly used dilution ratio is 1:4, which provides a balance between minimising the risk of non-target embolisation and maintaining an appropriate consistency for peripheral occlusion [28].

Typically, NBCA-to-Lipiodol dilution ratios range from 1:2 to 1:5 [18]. In their study, Kolu et al. used an even more diluted mixture (1:14) concluding that lower NBCA concentrations reduce the reflux by facilitating more distal occlusion [35]. This potentially lowers the risk of non-target embolisation. Baltacıoğlu et al. suggest a glue concentration of 12.5%, which may also reduce the risk of the microcatheter tip adhering to the glue cast [29]. Further research in this area could make NBCA more accessible for less-experienced operators by reducing the likelihood of severe complications.

Iodised oil also enhances fluoroscopic visibility due to its radiopaque nature [21,35]. This allows the interventionist to track the NBCA-Lipiodol mixture and accurately determine the moment of complete vessel occlusion. Due to the prompt polymerisation within a few seconds, which depends on the NBCA and iodised oil ratio, the procedure of glue embolisation takes less time than BAE using PVA particles [21]. As a result, radiation exposure for both the patient and the interventionist is reduced.

All these characteristics make NBCA a highly competitive embolic material, rivalling PVA particles and other embolisation agents.

### 5.3. Disadvantages

The main disadvantage of glue embolisation is the increased risk of severe complications such as non-target embolisation and tissue necrosis [3,5,14,18,27].

Unintentional occlusion of healthy vessels can result from NBCA’s retrograde flow. The most common causes of reflux include the anatomical position of the bronchial artery and technical errors, such as wedge positioning of the microcatheter, injecting an excessive volume of the mixture, or prematurely withdrawing the microcatheter [28,29]. The most concerning complication of glue embolisation is non-target embolisation of the spinal collaterals [3,28,36]. Its reported incidence in patients undergoing NBCA embolisation is approximately 0.71% [36]. However, spinal cord ischaemia is extremely rare, making it difficult to estimate precise statistics. Some studies include cases where patients exhibited neurological deficits that, while suggestive, do not definitively confirm spinal cord infarction [14,36].

Tissue necrosis can result from the occlusion of distal vascular beds, such as capillaries or vasa vasorum. It may affect the oesophagus, bronchi, or major blood vessels within the mediastinum [14,29]. However, this complication is infrequent, as the high viscosity of the mixture limits its spread beyond visibly observable vessel segments [29].

Other technical challenges include rapid polymerisation and accidental adhesion of the microcatheter tip to the NBCA mass [3,18,28].

Due to these complexities, NBCA embolisation requires significant experience from the interventionist [3,14,28]. Shamseldin et al. recommend that inexperienced operators first observe glue embolisation procedures performed by experienced endovascular specialists. Before performing the procedure independently, they should be supervised by an expert in the field. This approach has been effective in preventing major complications in clinical practice [28]. Training less experienced operators should initially be conducted in less complex clinical scenarios [5].

Despite these concerns, current research comparing NBCA to PVA in BAE has not demonstrated a higher incidence of complications following glue embolisation [14,21,27].

### 5.4. Clinical Use

Baltacıoğlu et al. detailed potential BAE complications, emphasising the need to prevent non-target embolisation [29]. Operators should adjust the NBCA-to-Lipiodol ratio to control polymerisation speed [5,35]. Kolu et al. suggested using diluted mixtures to prevent reflux, though overly low NBCA concentration can prolong polymerisation, risking distal bed occlusion, tissue necrosis, or catheter entrapment [35]. Optimising safe, effective NBCA–oil ratios remains a research priority. Shamseldin et al. emphasised proper microcatheter placement, confirmed by blood aspiration and radiopaque injection before embolisation [28]. Injections should be slow and continuous, forming a cast at the target vessel [29]. The catheter tip should be as close to the lesion as possible; wedge positioning may promote reflux [21,28,29]. If the catheter tip is stuck in the NBCA, the safest response is to leave it in place; forceful removal may damage the vessel [28,29]. NBCA outperforms other agents in haemoptysis control due to effective occlusion and low recurrence. Lee et al. showed NBCA’s superiority over PVA in primary lung cancer, as it occludes multiple small tumour-feeding vessels [21]. Nomori et al. observed tumour growth reduction in advanced, treatment-resistant cancers post-embolisation, possibly due to permanent blood supply disruption [32]. NBCA also functions independently of coagulation, making it ideal for anticoagulated patients [21].

Woo et al. found NBCA more effective than PVA in bronchiectasis, with no significant difference in aspergillosis or TB, likely due to ongoing neovascularisation and collateral recruitment [14]. Mazıcan et al. saw no significant NBCA advantage in life-threatening haemoptysis (mostly bronchiectasis-related), possibly due to a small sample. Recurrence was 3.3% (NBCA) vs. 17.9% (PVA/microspheres) [27]. Despite lacking statistical power, results still favoured NBCA. Recurrence mechanisms differed—recanalisation in the PVA/microsphere group vs. new vessel formation in the NBCA group—possibly influencing recurrence rates. Even diluted NBCA showed strong efficacy in massive haemoptysis, allowing more precise embolisation with less reflux, though comparisons to other agents or dilution levels remain lacking [35]. Glue embolisation is applicable in both bronchial and non-bronchial systemic arteries [5,14,21,27,29,35].

In summary, the current data show NBCA is as safe as other embolic agents when performed by experienced operators. However, broader studies are needed to refine technique, prevent complications, and expand its clinical use.

## 6. Gelatin Sponges (GS)

### 6.1. Short Description

Gelatin sponge (GS) is a biodegradable medical dressing used primarily to achieve haemostasis by applying it to a bleeding area. This non-calibrated particulate has been found to be effective and safe in medical practice. Historically, an embolic agent has been commonly used, though its occlusion level and in vivo travel trajectory persist unpredictably.

Gelatin-based systems are water-insoluble, flexible, sterile, and absorbable materials derived from porcine gelatin [26]. The embolic agents have potential to uptake large amounts of physiological solutions, the most common of which is blood. They also contribute to the initiation of clotting by providing a mechanical matrix. GS acts as a scaffold to promote tissue regeneration due to cell adhesion. A porous structure remains flawless during immersion in fluids [38].

Additionally, the material can be provided in a fluid form or “slurry” which consists of tiny hand-cut cubes mixed with contrast medium and saline. Some studies advocate the use of this structure to allow smaller cubes to lodge more distally in smaller-calibre vessels. Hawthorn et al. determined the effects of slurry on postoperative morbidity of women undergoing caesarean section. In the authors’ practice, the fluid form of GS is used frequently for embolisation in cases of obstetric haemorrhage and is beneficial and safe in their experience [39].

### 6.2. Advantages

One of the significant benefits of gelatin sponges is their high efficiency in controlling bleeding by initiating platelet adhesion, activation, and aggregation. This form can promote coagulation due to its loose porous architecture which allows it to absorb water, thus making the blood sticky. Haemostasis is generally achieved within 5–10 min [40]. Excellent control in capillary, arteriolar, and venous bleeding is because of its absorption capacity, which is roughly 45 times its weight. Compatibility of GS with the human body protects against the possibility of severe immune reactions. This non-calibrated particulate is obtained from collagen in pigs and has a neutral pH. Furthermore, the risk of permanent foreign-body reactions is low due to the GS’s resorption over time [38]. It is also widely available in many clinical faculties due to its low cost and safety. The affordable effect is especially noticeable in resource-limited regions. Compared to other embolic tools such as coils or PVA, this gelatin form provides economical choice without the need for specialised handling.

Moreover, the form is shapeable, which means it can be easily modified to the desired sizes to fit unique surgical needs. Particulate dimensions are commonly in the range of 0.5–2 mm. GS also can be managed in different forms, like slurry, pledges, or torpedoes [26]. Another key advantage is biodegradability; the human body can absorb the gelatin-based system within 4–6 weeks. In practice, this is useful in cases where conventional alternatives or ligation are ineffective and in situations where the non-absorbable forms are not recommended [38,41]. This haemostatic agent is available for temporary embolisation procedures without leading to permanent vessel blockage or ischaemia. Its short-term nature allows repeat procedures if rebleeding occurs, even if it could be the same vessel. This eventuality is beneficial for patients with insufficient initial occlusion or evolving disease conditions. Chronic vessel closure may limit healing options in the future, so GS does not lead to this case. Nagano et al. focused their study on using a gelatin sponge as the main tool for the treatment of cryptogenic haemoptysis. Their results show that recanalisation occurs in several weeks to months. GS can terminate haemorrhage and suppress serious rebleeding with the slightest complications in patients. Research demonstrates that the recurrence-free rate stays up to 87.4% during a 2-year monitoring phase. Additionally, life-threatening pulmonary haemorrhage does not occur at recurrence [41]. The performance of gelatin sponge should be comparable to long-lasting alternatives such as PVA, coils, or NBCA [18].

### 6.3. Disadvantages

Despite many different benefits, GS has some disadvantages. The degradable nature may result in early restoration of blood flow or, on the other hand, long-term blockage [26]. Vessel recanalisation within weeks or months could lead to the need to repeat the therapeutic process again [41]. Due to its unpredictable degradation profile, GS may not be the primary choice in cases requiring permanent occlusion such as recurrent haemorrhage disorders, progressive pulmonary diseases, or vascular malformations [3]. Furthermore, in BAE for haemoptysis, a gelatin-based system has been noted to have lower potency compared to other haemostatic materials, like cyanoacrylate-based substances or PVA. Shimohira et al. observed poor haemostatic effectiveness and higher likelihood of recurrence in patients with pulmonary aspergilloma [42]. Giammalva et al. reported that in neurosurgical applications, the gelatin form must be precisely managed because of its swelling effects. These properties could cause compression of juxtaposed structures [38]. Another disadvantage is the fact that GS can increase injury infection, principally in contaminated wounds. Moreover, embolisation needs careful choices of haemostatic agents’ size to stay away from complications like infractions. Gelatin materials of diameters smaller than 325 μm may pass through inadvertent vascular pathways or bronchopulmonary anastomoses [42]. Additionally, the adhesion between the embolic tool and vessel wall is not strong enough and is easy to peel off. GS only stops bleeding when the human body has a functioning clotting system. In this case, materials derived from porcine gelatin are not preferred for patients with impaired coagulation mechanisms. Some authors report that the use of mixture with glutaraldehyde for the interaction can reduce solubility. It may help with tissue adhesion in the liver, in gastrointestinal surgery, etc. Unfortunately, it contains aldehydes, which have mutagenicity and carcinogenic potential [40].

### 6.4. Clinical Use

Gelatin sponge is widely and effectively used in several medical sectors caused by its embolisation and haemostatic properties. This method has been applied in neurosurgical procedures, anorectal surgeries, nasal bleeding treatments, urological interventions and numerous others. In interventional radiology, it often controls excessive bleeding, such as in cases of haemoptysis [3]. Ishikawa et al. noticed that, in Japan, 79% of BAE for cryptogenic haemoptysis utilised this form of gelatin substance [36]. In surgical procedures, the method is utilised to manage bleeding in spaces where traditional procedures like ligation or sutures are impractical [38]. Nevertheless, its temporary result means that it is not always the favoured choice for embolisation, particularly in cases demanding long-term vessel occlusion. Despite its limitations, it continues to be generally utilised in surgical and interventional faculties due to its biodegradability, cost-effectiveness, and haemostatic capabilities.

Gelatin sponge has been a valuable embolic agent since 1973. Initially, this material was commonly used in therapy; then, modifications were gradually introduced. Panda et al. observed that the major difference in studies conducted before and after 2010 is the higher frequency of using selective catheterisation to decrease complications. Additionally, this study reported the increased usage of alternative embolic products such as polyvinyl alcohol particles or glue instead of GS [3,43].

To summarise, gelatin sponge is a widely available, safe, relatively inexpensive embolic agent with decades of clinical experience. Nevertheless, it usually serves as a temporary solution and is less effective in patients with impaired coagulation.

## 7. Coils

### 7.1. Short Description

Catheter-provided embolisation plays a crucial role in interventional radiology. Coils successfully treat pathologies from head to toe. Their wide utility makes them one of the most important devices in a variety of clinical scenarios [44]. Coils can be used alone or in combination with other haemostatic alternatives such as gelatin sponges or PVA to induce thrombosis and permanently block blood vessels. Generally, this embolic agent works in a totally different mechanism than other bloodstream-dependent haemostatic tools such as GS, PVA, or NBCA. This metallic structure provides a more proximal blockage compared to liquid materials [3,16,36].

Of all the coil properties, biocompatibility is considered before fabrication. Primarily, coils are made from metal material such as platinum or steel. To reach optimal flexibility and strength, metals are often developed as alloys [45]. Once a metal is chosen, the endovascular embolisation tool is created by a series of transformations. The primary form is fabricated in linear structure with a range of diameter from 0.00175 to 0.003 inch. This central factor regulates coil stiffness. The secondary formation is wounding the wire around a mandrel. The number of turns per unit of length around the mandrel represents the second diameter between 0.010 and 0.015 inch. Finally, the tertiary structure has several different shapes like spherical, helical or complex. This configuration also is developed with characteristics diameter and length. Coils are commonly packed as 3 mm (tertiary diameter) × 4 cm (length). Each metal material or configuration is designed to maximise various vessel occlusion [45].

### 7.2. Advantages

Biocompatibility is one of the most important properties. This factor allows successful treatment without triggering adverse systemic responses. Coils come in a variety of sizes, shapes, deployment methods and coating structures, making them suitable for a wide range of procedures. Size is commonly designated in three numbers; the first number is the wire’s diameter (0.008 to 0.052 inches), the second is the length (1 to 300 mm) and the third is the diameter of the coil complex (1 to 27 mm). Shapes include straight, conical, helical, tornado, S-shaped, J-shaped, C-shaped, and three-dimensional shapes. Typically, coils should be sized 20–30% larger than the size of the goal vessel in case of migration and unintended distal embolisation. This rule helps in placing the agent in the desired vessel location. Catheter-provided structures are available in a wide range of metallic materials. Examples of metals include nitinol, platinum, nickel, iridium, and tungsten. A generally used alloy is platinum (92%) mixed with tungsten (8%). This combination offers a balance of mechanical durability and fluoroscopic visibility [44].

Another key advantage is permanent occlusion. Coils indicate a long-lasting blockage with a low recanalisation rate. Endovascular embolisation has high thrombogenicity, which means the process of forming thrombi is instantly achieved [3,6].

This haemostatic agent shows effectiveness in severe haemorrhage. Dohna et al. focused on short-term and long-term outcomes of patients with CF and haemoptysis after BAE using metallic coils alone. During a 5-year observation interval, authors reported an 8.8% recanalisation rate. Super-selective bronchial artery coil embolisation (ssBACE) is less radical. This method saves smaller vessels to supply the lung tissue to stay lively. It also helps antibiotic therapy with accession to inflamed territories. ssBACE considerably improves functional end-expiratory volume in 1 s in % predicted (FEV1% pred.). Dohna et al. noticed an increased FEV1% pred. in all patients from 45.7% to 49.1% and from 26.3% to 31.5% in severely compromised patients (with FEV1% pred. <40%) [46,47]. Other observational studies have shown that the 1-year haemoptysis-free survival rate using ssBACE is 87%. Ryuge et al. described four possible ways of recurrent haemoptysis. Authors demonstrate the development of new haemoptysis-related arteries, bridging conventional collaterals and recanalisation. The mechanism of the process was considerably different among numerous subsets of patients [48].

Ishikawa et al. observed that hospital interventions including oxygenation, mechanical ventilation, and transfusion were lower in the coils group. Catheter-provided material would prevent future repeat BAEs due to embolisation at the intended position and not cause reflux [36].

### 7.3. Disadvantages

Coils, compared with other embolisation alternatives, have a higher length of hospitalisation and total health care cost [36]. ssBACE requires local anaesthesia at the injection site and skilled interventionists. In contact with blood, coils instantly form thrombi, achieving excellent haemostasis. Nevertheless, using this embolic agent can cause some problems for inexperienced operators [3,6]. In the first place, due to the drastic reduction in regional blood volume and underlying pathological erosion, the targeted haemorrhaging vessel becomes significantly smaller and distorted, making it difficult to use metallic material. Moreover, concerns have been raised regarding the blockage of the proximal vessel following the use of super-selective embolisation tool placement. It could cause some possible future embolisation complications of the distal part of the same vessel in cases of recurrent haemorrhage [47]. Additionally, metallic coil is a widely used agent in haemoptysis therapy, but its permanent effects cause difficulty in recycling [6].

### 7.4. Clinical Use

Metallic coils are widely used in treating diverse clinical conditions due to their customizability. Catheter-based embolisation plays a key role in managing AVMs and life-threatening bleeding [3,44]. Coils are employed in conditions like pelvic congestion, varices, lymphatic leaks, BPH, uterine fibroids, and cancer. In oncology, coils aid in tumour devascularisation [44].

When combined with other agents, coils effectively occlude pseudoaneurysms and non-bronchial systemic collaterals [3]. Their rapid haemostatic action is also suitable for trauma-related bleeding [6]. In interventional radiology, ssBACE using coils is effective for massive haemoptysis, especially in CF cases [46].

Ishikawa et al. reported coils as safe and effective for permanent control of severe haemoptysis in CF, including emergency settings [47]. Fu et al. also demonstrated coil success (94.9%) in rare cases like bronchial haemoptysis from intercostal pulmonary venous shunts [16].

While agent selection depends on availability and operator preference, many authors endorse coils as a first-line option for BAE [46,47,48]. Their adaptability in size/shape enhances clinical flexibility. Coils’ technical properties are essential to optimising outcomes in haemostatic interventions [44].

To conclude, coils offer permanent vascular occlusion, particularly within proximal vessels. Their efficacy makes them a valuable embolic agent in the management of AVMs and severe haemoptysis.

The key characteristics of embolic materials discussed in this review are summarised in Table 1.

## 8. Practical Implications and Decision Algorithm

### 8.1. Clinical and Anatomical Considerations in Embolic Agent Selection

The choice of embolic material should not rely solely on operator experience but must take into account both clinical context and vascular anatomy at the target site. Clinical context is crucial, as the goals of embolisation may differ: in some cases, it is performed as a temporary, palliative measure (e.g., in patients awaiting transplantation or with poor prognosis), whereas in others it is intended as a definitive intervention (e.g., tumour-related neovascularisation).

Because common aetiologies of haemoptysis (bronchiectasis, tuberculosis and its sequelae, cystic fibrosis, aspergillosis, malignancy) often coexist and share pathophysiological mechanisms, recommendations cannot be based on aetiology alone. Instead, decision-making should be individualised, integrating patient condition, vessel characteristics, and anticipated durability of occlusion.

Multidisciplinary evaluation, including assessment of the underlying disease by the pulmonologist and vascular anatomy by the interventional radiologist, is therefore essential. Pre-procedural MDCT can provide valuable guidance [53].

The key questions guiding the choice of embolic agent are illustrated in the decision tree (Figure 2).

### 8.2. Recurrence Risk and Material-Specific Outcome

Clinical assessment and evaluation of the underlying mechanism of haemoptysis are essential for estimating recurrence risk and selecting the most appropriate embolic material. Recurrence risk assessment is central to treatment planning, as major causes include incomplete embolisation, recanalisation (especially with biodegradable agents such as gelatin sponge), neovascularisation driven by inflammation or tumour-related factors, and recruitment of collateral circulation. Furthermore, mechanisms of recurrence arise not only from the intrinsic properties of embolic materials, but also from procedural factors such as patient selection, accurate identification of culprit vessels, and the presence of collateral circulation [5,17,28,42,46,48,52].

The recurrence spectrum is relatively broad for all embolic agents analysed. The lowest recurrence rates are reported after NBCA use, which aligns with previous studies and the general consensus regarding the efficacy of glue embolisation. Moreover, NBCA consistently demonstrated the highest clinical success rate across the included studies (see Table A3). However, this material is also associated with a relatively high complication rate (14.9%). Despite differences in physical properties between microspheres and PVA, both demonstrate similar average recurrence rates. PVA is the most extensively studied agent, with the broadest range of recurrence values, likely due to its use across nearly all aetiologies, which makes it difficult to assess its efficacy independently of clinical context. Only a few studies included in this review used gelatin sponge as the sole embolic material, making it difficult to draw firm conclusions. However, its transient effect and tendency toward recanalization have been confirmed in multiple reports. Despite its limited durability, gelatin sponge is well tolerated clinically, as evidenced by the lowest complication rate (3.3%) among the analysed agents. Table 1 provides aggregated data on recurrence and clinical success rates for each embolic agent, as reported in the included studies.

Given the heterogeneity of follow-up duration, aetiologies, and clinical scenarios, direct standardisation of recurrence values within this scoping review was not feasible, and conclusions must be drawn with caution.

### 8.3. Emerging Technologies and Materials in Bronchial Artery Embolisation

Endovascular interventions, including BAE, represent a promising direction in modern medicine due to their minimally invasive nature and considerable therapeutic potential, which is particularly important for patients ineligible for surgery or requiring urgent intervention. Although this review focuses on the primary goal of BAE, namely vessel occlusion and haemoptysis cessation, it is worth emphasising the broader possibilities of embolisation.

Microspheres are considered an alternative to PVA particles by CIRSE, offering comparable embolic properties [30]. However, advances in microsphere manufacturing allow for additional functionalities. One such innovation is the development of drug-eluting beads (DEBs), mentioned in Section 4.2. Advantages dedicated to microspheres, which combine cytotoxic drug delivery with ischaemia induced by vessel occlusion. Nezami et al. demonstrated the safety and promising response of pulmonary metastatic tumours to DEB-BACE [54].

Xiaobing et al. explored bronchial arterial infusion chemoembolisation (BAICE) for primary lung cancer, combining intra-arterial chemotherapy with embolisation using PVA or gelatin sponge. The cytotoxic properties of the drugs reduce neovascularisation and increase embolisation efficacy, while vessel closure enhances local drug concentration in tumour cells by reducing washout [55].

Compared to BAICE, DEBs offer more precise drug delivery, prolonged therapeutic effect, and controlled dosing, potentially reducing systemic toxicity.

The decision algorithm proposed in this review includes biodegradable microspheres as an alternative to GS in patients with coagulopathy. These materials, derived from biopolymers and synthetic polymers, offer excellent biocompatibility and adjustable degradation time, ranging from 6 h (Hydrogel Microspheres with Hydroxylamine-Containing Crosslinker) up to 6 months (Hydrophobic Linear Polymer-Based Microspheres), allowing tailored recanalisation timing [26].

Since detailed discussion of all materials exceeds the scope of this review, readers are referred to the work of Hu et al. for further insights.

In addition to embolic agents, imaging technologies continue to evolve. Cone-Beam Computed Tomography (CBCT) enables three-dimensional visualisation of vascular pathways, pulmonary parenchyma, and other tissues. However, it is associated with increased patient exposure to radiation and the need for a larger volume of administered contrast [56].

These developments illustrate the ongoing evolution of embolic materials and imaging tools, which may significantly influence future standards of BAE practice.

**Figure 2 arm-93-00035-f002:**
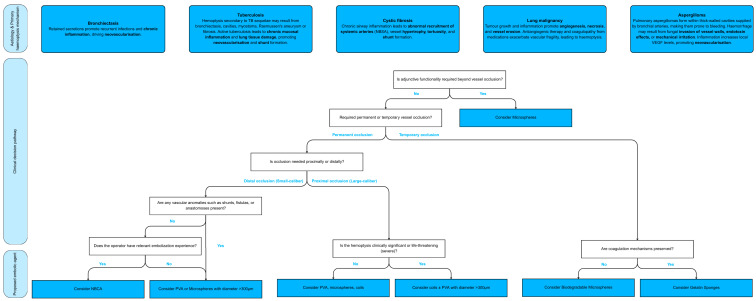
Decision flowchart for embolic agent selection in bronchial artery embolisation. Polyvinyl alcohol (PVA), N-butyl-2-cyanoacrylate (NBCA), Gelatin sponges (GS), Bronchial artery embolisation (BAE), Tuberculosis (TB), Cystic fibrosis (CF), NBSA (Non-bronchial systemic arteries), Vascular endothelial growth factor (VEGF). Aetiology and primary haemoptysis mechanism references: [2,10,42,46,57,58,59].

## 9. Discussion

This scoping review identified key factors influencing embolic agent selection in haemoptysis management, including clinical context, vascular anatomy, and recurrence risk. Due to variability in study designs, definitions of clinical success, and follow-up duration, direct comparison of outcomes across embolic agents remains challenging. NBCA consistently showed favourable outcomes, including reduced recurrence and the most effective immediate haemoptysis control, while PVA and microspheres showed comparable outcomes. Gelatin sponge, though widely used, was associated with higher recurrence due to its biodegradable nature. Coils, offering permanent occlusion, are especially valuable in proximal embolisation and severe presentations.

Bronchial artery embolisation was used for the first time as a treatment for haemoptysis in 1973 [43]. Since then, the technique of BAE has evolved, and also embolic materials have changed. Formerly, gelatin sponges were commonly used due to their availability and low cost. However, as they are reabsorbed over time, their use can lead to recanalisation and recurrence of haemoptysis [42]. Today, non-resorbable embolic materials, such as coils, polyvinyl alcohol, and liquid embolising agents are preferred [3]. However, no agreement exists on which agent is most appropriate for specific clinical circumstances. This review explores the evolving role of bronchial artery embolisation in haemoptysis management, emphasising the selection and effectiveness of embolic materials, strategies to minimise recurrence, and its diverse applications, from emergency interventions for massive haemoptysis to palliative care aimed at enhancing patient quality of life. The analysis of selected studies containing precise data on the embolisation agents used (Figure 3) reveals that the choice of specific agents may depend on multiple factors.

PVA particles were used in all included studies, confirming their well-established role in the BAE procedure. Coils were also frequently used, often as a complementary method when initial embolisation failed to achieve full bleeding control. A notable observation is the growing popularity of gelatin sponges in recent years, despite their well-known limitations (Figure 4). This trend is likely driven by their low cost and validation in numerous studies. Regional differences in embolic agent selection appear to be influenced not only by clinical preference but also by reimbursement policies and material availability. For instance, gelatin sponges are predominantly used in Asia. In Japan, PVA particles and embospheres are not reimbursed by the national health insurance system, which may explain the substitution with gelatin sponges as a temporary particulate agent [36]. Such systemic factors may provide explanation for variations in practice across regions and should be considered when interpreting comparative outcomes. Microspheres of various sizes were reported in a limited number of studies. Notably, Lin et al. used them as the dominant embolic agent [62], which may indicate that microspheres remain a niche option, typically chosen by clinicians familiar with their handling. Glue agents (e.g., NBCA) were also occasionally applied during BAE despite their permanent nature, which may illustrate the need for skilled interventionists. Other embolic substances (vascular plug, covered stent, thread) were used intermittently in small quantities.

It is worth noting that many of the analysed studies summarise outcomes of procedures performed over extended periods, which complicates the assessment of temporal trends. Nonetheless, it can be inferred that a key factor influencing the choice of intervention materials is the personal experience of the interventionist and local practice patterns. Bronchial artery embolisation appears to be a valuable treatment option for haemoptysis across various clinical contexts. The findings presented above underscore the need to standardise BAE procedure recommendations. According to CIRSE guidelines from 2022, PVA particles with a diameter of 355–500 μm are currently recommended. However, there is still great heterogeneity of embolising agent usage in research [30].

This review maps the current landscape of embolic agent use in BAE and highlights the need for standardised guidelines and further comparative studies to inform material selection.

## 10. Limitations

Several limitations of this review should be considered when interpreting the results.

This article is structured as a scoping review, aiming to map the available literature rather than to provide a full critical appraisal of the included studies or a quantitative synthesis of the findings. Consequently, the conclusions should be interpreted as an overview of the existing evidence rather than as a definitive assessment of the efficacy of specific embolisation methods or materials.

The literature search was conducted exclusively in the PubMed database, which may have led to the omission of relevant publications indexed elsewhere. Furthermore, only articles published in English were included, introducing potential language bias and limiting the comprehensiveness of the evidence base.

Study selection and data extraction were performed primarily by a single reviewer; although uncertainties were discussed with a second researcher, this approach carries a risk of subjectivity in study selection and data interpretation. Furthermore, the absence of formal protocol registration reduces transparency and reproducibility.

The inclusion criteria restricted the review to publications from 2019 to 2024, ensuring the focus was on recent evidence, but this may have resulted in the exclusion of older yet potentially valuable studies.

An additional limitation lies in the heterogeneity of study designs, including differences in patient populations, embolic materials applied, definitions of complication and clinical success, and follow-up duration. By aggregating major and minor complications into a single composite rate, we cannot distinguish between severe and transient events, and inconsistent reporting of minor complications (e.g., transient chest pain) may have led to over- or underestimation of overall complication rates. Refraining from reclassifying haemoptysis severity based on volume thresholds limits our ability to assess embolic agent performance by bleeding intensity.

## 11. Conclusions

The procedure of embolisation has been used in medicine for more than 50 years, during which it has undergone significant development and improvement. Today, there is a wide range of embolisation agents, and diversity is constantly growing. The use of embolisation in various clinical contexts is also steadily increasing. This progress brings new challenges in weighing the benefits to the patient against the potential risks of complications. The rapid evolution of implemented techniques and a lack of universal guidelines make the choice of embolising agent largely dependent on the interventionist’s experience.

Bronchial artery embolisation serves as a therapeutic option for haemoptysis in both clinically stable patients seeking relief from chronic symptoms and in those with life-threatening presentations requiring immediate care. This implies that BAE should be a flexible method, tailored to the individual patient.

Despite a strong evidence base, further research is needed on BAE materials and the formulation of specific guidelines regarding their use, depending on clinical indications, including haemoptysis severity, patient prognosis, and operating conditions (e.g., diameter of embolised vessels, presence of AVMs).

## Figures and Tables

**Figure 1 arm-93-00035-f001:**
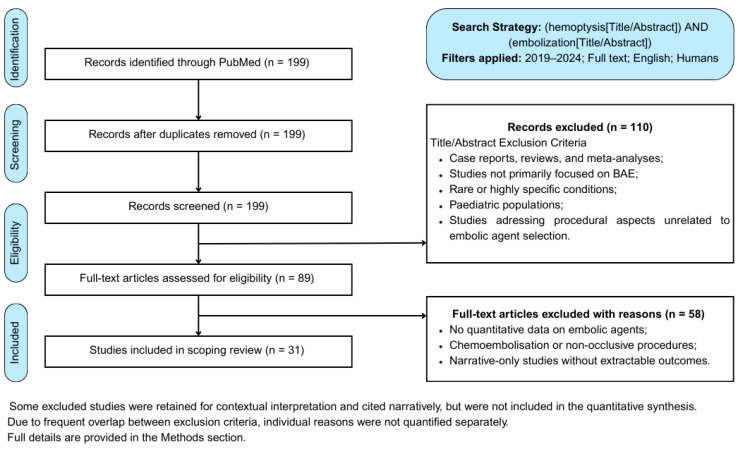
PRISMA ScR flow diagram of the study selection process (adapted from Peters et al. [12]).

**Figure 3 arm-93-00035-f003:**
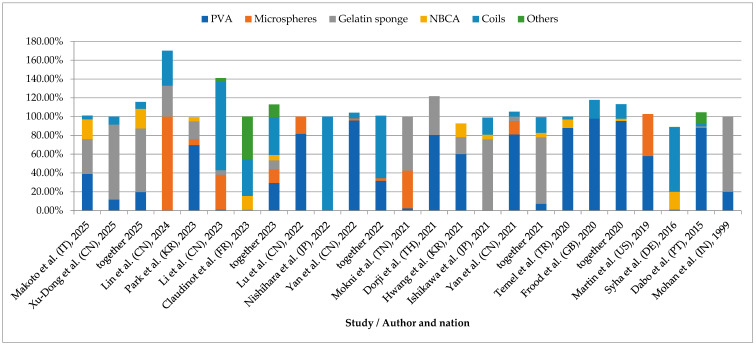
Comparison of embolic materials utilised in BAE: literature overview. In some studies, more than one embolic material was used in the same procedure, resulting in cumulative percentages exceeding 100%. Based on data extracted from selected studies [4,8,10,25,36,51,59,60,61,62,63,64,65,66,67,68,69,70,71,72]. Polyvinyl alcohol (PVA), N-butyl-2-cyanoacrylate (NBCA).

**Figure 4 arm-93-00035-f004:**
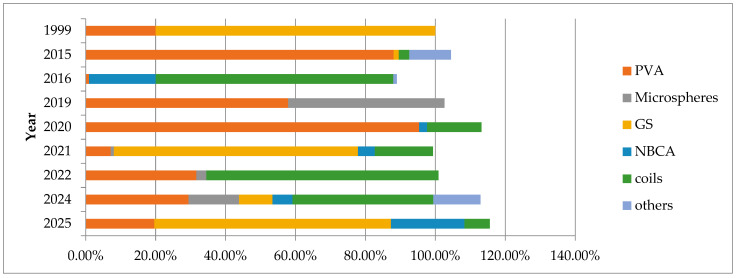
Yearly distribution of embolic materials used in bronchial artery embolisation. In some studies, more than one embolic material was used in the same procedure, which results in cumulative percentages exceeding 100%. Based on data extracted from selected studies [4,8,10,25,36,51,59,60,61,62,63,64,65,66,67,68,69,70,71,72]. Polyvinyl alcohol (PVA), Gelatin sponges (GS), N-butyl-2-cyanoacrylate (NBCA)**.**

**Table 1 arm-93-00035-t001:** Comparison of main embolic agents.

	PVA	Microspheres	NBCA	GS	Coils
Group of embolic agents	Particulate—irregular; 300–500 µm is most often selected	Particulate—standardised size range of 100–1300 μm and spherical shape	Liquid	Particulate—irregular; the range of 0.5–2 mm is commonly used	Mechanical Occlusion Devices—irregular; range of size is commonly designated in three numbers
Mechanism of action	When released into the bloodstream from a microcatheter, particles expand and occlude the targeted vessel	Mechanical occlusion of vessels, conforming to their size	Polymerises upon contact with tissues, forming a permanent occlusion	Uptake large amounts of physiological fluids, initiation of clotting by providing a mechanical matrix. GS acts as a scaffold to promote cell adhesion and tissue regeneration	Catheter-provided embolisation, in contact with blood instantly forms thrombi, achieving excellent haemostasis
Main advantages	Low cost; availability; customizable fit; well-documented safety in BAE	Aggregation-resistant; compressible; wide range of sizes; low cost	Low recanalisation rate; short time of procedure; radiopaque; more distal-embolisation	Widely available; relatively inexpensive; safe; easily modified to the desired sizes and accumulated years of medical use; no impact on haemodynamics	Wide range of sizes, shapes, deployment methods and coating structures; fluoroscopic visibility; safe; used independently or in combination with other embolisation agents
Main disadvantages	Poor adhesion to the endothelium; Aggregation within the catheter; Radiolucent	Radiolucent; Immunogenic	Requires significant experience; Increased risk of non-target embolisation and tissue necrosis	Degradable nature, poor haemostatic effectiveness in patients with pulmonary aspergilloma and increase injury infection	Higher length of hospitalisation and total health care cost; Requires experienced interventionists
Control over embolisation	Low—size selection only; irregular shape may lead to unpredictable aggregation	High—uniform size and shape allow better control	Moderate—NBCA-Lipidol ratio determines polymerisation speed	Moderate—wide range of size, impermanence of the process	Moderate—customizable fit
Ability to embolise small vessels	Limited-aggregation may block larger vessels	High-size selection allows embolisation of both small and large vessels	High-reaches the most distal vessels	High-customizable size can be used in smaller calibre vessels	Limited-metallic structure provides a more proximal blockage
Risk of reflux/non-target embolisation	Moderate-mainly when particle sizes are below 300 µm or in the presence of shunts	Moderate-mainly when particle sizes are below 300 µm, in the presence of shunts and after adding a contrast agent	High-especially due to technical errors such as wedge positioning of the microcatheter, injecting an excessive volume of the mixture, or prematurely withdrawing the microcatheter	Moderate-diameters smaller than 300 μm may pass through	Low-due to embolisation at the intended position
Use in BAE	Frequently used in standard cases (e.g., bronchiectasis)	Haemoptysis associated with malignancies; Other similar to PVA	Haemoptysis associated with malignancies; Effective in massive haemoptysis; Efficacy similar to particulate agents in chronic, progressive diseases	Mostly for cryptogenic haemoptysis; Severe persistent haemoptysis	Used for massive haemoptysis, especially in CF, aneurysms, AVMs
Durability of effect	Permanent, although recanalisation frequently occur	Permanent or temporary (depend of material biodegradability)	Permanent embolisation	Temporary embolisation	Permanent occlusion
Additional functions	No additional functions	Might contain medical preparations or radioactive isotopes	May help prevent tumour growth by permanent cutting off blood supply	No additional function	No additional functions
Possible complications	Premature embolisation due to aggregation within the catheter; Non-target embolisation;	Non-target embolisation; Foreign-body reactions	Non-target embolisation; Risk of catheter adhesion; Tissue necrosis	Non-target embolisation; Risk of compression of approximal structures; Increase wound infection	Complicate the embolisation of the distal part of the same vessel; Increase wound infection; Unintended embolisation
Aggregated Outcomes Across Included Studies [4,5,9,15,17,22,23,27,31,32,35,42,46,49,50,51] ^1^					
Average Complications Rate	23.7%	28.2%	14.9%	3.25%	N/A
Average Clinical Success	96.9%	95.9%	97.9%	97.5%	94.9%
Recurrence Rates (Range)	8.9–81.9%	6.3–61.6%	3.3–9.5%	17.4–26.9%	8.4–51.2%
Average Recurrence Rate	36.2%	25.9%	7.8%	22.2%	31.6%
Mechanisms of Recurrence—Based on Selected Individual Studies					
Leading Recurrence Mechanisms	Incomplete embolisation Disease progression Collateral recruitment Premature embolisation Recanalisation due to lack of adhesion [17,19,31]	Incomplete embolisation Disease progression Collateral recruitment [17,31]	Incomplete identification of HRAs [5,28]	Recanalisation due to resorption [52]	Recanalisation due to coil compaction Incomplete occlusion Collateral recruitment [46,48]

Abbreviations: Polyvinyl alcohol (PVA), N-butyl-2-cyanoacrylate (NBCA), Gelatin sponge (GS), Bronchial artery embolisation (BAE), Arteriovenous malformations (AVMs), Cystic fibrosis (CF), Haemoptysis-related arteries (HRAs), Non-applicable (N/A). ^1^ All values were extracted from the included studies that reported these specific outcomes for each embolic material. Detailed individual study values are provided in Appendix A (Table A3).

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
