# Peer review of "Bronchial Artery Embolisation in Haemoptysis Management: A Scoping Review with Emphasis on Embolic Materials and Indications"

_arm, 2025, doi:10.3390/arm93050035_

Round 1
Reviewer 1 Report
Comments and Suggestions for Authors
Dear authors,
Although this is an interesting topic and represents a great amount of work done, I have some concerns about the methodology of this review.
The research question and objectives are not well defined.
The methodology of your work, including search terms, inclusion/exclusion criteria, the exact number of included studies, and many important details, is not described.
The discussion is not standardly presented. It would be better to start your conversation with a summary of the key findings.
I recommend extensively rearranging your manuscript to better section different parts of your work.
Author Response
A collective response to the reviewers' comments can be found in the attached file.

Reviewer 2 Report
Comments and Suggestions for Authors
The paper is a scoping review on bronchial artery embolization (BAE) for haemoptysis. It compares commonly used embolic agents (PVA, microspheres, gelatin sponge, NBCA, coils), notes variable practice and outcomes, and calls for clearer guidance.
Main recommendations to improve quality:
- Make the review process transparent
Add a short “Methods” box: databases searched, dates covered, inclusion/exclusion, how studies were selected and charted (a PRISMA-ScR style flow would help). This will address heterogeneity concerns and allow readers to judge evidence strength.
- Standardise key definitions up front
Define “massive” vs “non-massive” haemoptysis (report the range in the literature and state what you adopt for the review). This resolves interpretation issues later in the text.
- Be specific about practice implications
Turn narrative points into “when to use what” guidance (e.g., NBCA for distal/tumour feeders in experienced hands; coils for durable proximal control, often combined with particles; gelatin sponge as a temporary or cost-driven option). Support each bullet with one or two numbers if available.
- Quantify and discuss recurrence more coherently
Gather recurrence timing and mechanisms by agent in one place (e.g., PVA: early recurrence via recanalisation/incomplete embolisation). This makes the take-home message clearer.
- Highlight regional and access differences
Briefly note how reimbursement and availability shape material selection (e.g., higher gelatin sponge use in Japan due to reimbursement), so readers understand why practice varies.
Author Response

(The authors gave the same response as above.)

Reviewer 3 Report
Comments and Suggestions for Authors
This manuscript provides a comprehensive scoping review of bronchial artery embolisation (BAE) as a treatment for haemoptysis, with a particular focus on the variety of embolic materials used, including polyvinyl alcohol (PVA), microspheres, N-butyl-2-cyanoacrylate (NBCA), gelatin sponges, and coils. The review synthesizes recent literature to discuss the advantages, limitations, and clinical applications of these embolic agents, highlighting their roles in different clinical scenarios and the need for tailored decision-making. The manuscript also underscores the lack of standardized guidelines for embolic material selection and calls for further research to optimize clinical outcomes. The review is well-structured and covers a broad spectrum of embolic materials, providing detailed descriptions of their properties, clinical uses, and associated complications. Additionally, the manuscript effectively integrates recent studies, offering updated insights into the efficacy and safety profiles of each embolic agent. However, I recommend this manuscript for publication after major revisions addressing the points below.
Points for Consideration / Suggestions:
- Depth of Comparative Analysis: While the manuscript describes each embolic material extensively, a more direct comparative synthesis or tabulated summary of their clinical efficacy, complication rates, and recurrence outcomes could enhance reader comprehension and practical utility.
- Guideline Development: The manuscript highlights the lack of standardized guidelines but could benefit from a more detailed discussion on potential pathways or frameworks for developing such guidelines, possibly incorporating multidisciplinary perspectives.
- Emerging Technologies and Materials: Consider including a brief section on emerging embolic materials or novel techniques in BAE that may impact future practice, to provide a forward-looking view.
- Limitations of the Review: A clear statement on the limitations of the scoping review methodology, such as potential publication bias or heterogeneity in study designs, would strengthen the manuscript’s rigor.
- Figures and Tables: Inclusion of schematic diagrams or flowcharts summarizing embolic agent selection based on clinical scenarios could improve the manuscript’s educational value.
Author Response

(The authors gave the same response as above.)

Round 2
Reviewer 1 Report
Comments and Suggestions for Authors
The authors have addressed the issues well. However, the PRISMA flowchart is still incomplete. Please provide a more detailed version of the flowchart, including the reasons for exclusion and the number of references obtained from each database.
Author Response
The Flow Diagram has been revised to include a detailed search strategy, specific exclusion criteria at both the screening and full-text stages. The review was based on a single database (PubMed), which is now clearly indicated to reflect the scope and limitations of the search. Due to frequent overlap between exclusion reasons, individual categories were not quantified separately, as explained in the Methods section. We have also clarified that some excluded studies were cited narratively to enrich the contextual understanding of embolisation in haemoptysis. Thank you for the positive review.
Reviewer 2 Report
Comments and Suggestions for Authors
My previous comments have been visited by the authors and the manuscript can be accepted in its current form.
Author Response
Thank you for the positive review.
Reviewer 3 Report
Comments and Suggestions for Authors
The authors have addressed my questions.
Author Response
Thank you for the positive review.